# When the Past Teaches the Future: Earthquake and Tsunami Risk Reduction through Episodes of Situated Learning (ESL)

**Giovanna Lucia Piangiamore** [1,*] and **Alessandra Maramai** [2]

1   Istituto Nazionale di Geofisica e Vulcanologia, Sezione Roma2, Sede di Lerici, 19032 Lerici, Italy
2   Istituto Nazionale di Geofisica e Vulcanologia, Sezione Roma1, Sede di Roma, 00143 Rome, Italy; alessandra.maramai@ingv.it
*   Correspondence: giovanna.piangiamore@ingv.it

**Abstract:** The past offers important lessons with regard to facing the future with greater awareness. In this context, school plays a key role in spreading knowledge of natural phenomena and in promoting behavior change. Together with researchers, teachers can be strong allies to build more resilient future citizens. The Istituto Nazionale di Geofisica e Vulcanologia (INGV) school training activities provide tools to prepare for the next earthquake and/or tsunami. Approximately 5000 students, from both middle schools (ISCED 2) and high schools (ISCED 3), were involved in *active learning* activities based on a flipped-up approach during specific online scientific events during the pandemic. Online lab activities were conducted during European Researchers' Night ("*Earthquakes: history teaches us the future: researchers for a day with experimentation in didactics for ESL*") and during both World Water Day 2021 and World Earth Day 2021 ("*Tsunamis: history teaches us the future researchers for a day with experimentation in didactics for ESL*"). These two *Episodes of Situated Learning (ESL)* experiences triggered students' interest, favoring remote learning, developing life skills, and focusing on historical seismic studies of both past earthquakes and tsunamis.

**Keywords:** *Episodes of Situated Learning (ESL)*; *active learning*; *critical thinking*; prevention; natural risk preparedness; damage; earthquake; tsunami; seismic risk reduction; education





## 1. Introduction

As Thucydides taught as early as the 5th century B.C., "*One must know the past in order to understand the present and affect the future*". The "past teaches the future" concept stands at the basis of Uniformitarianism, formulated by the geologists James Hutton and Charles Lyell, who summarized their understandings with the well-known slogan, "the present is the key to the past". More recently, Carl Sagan famously wrote, "*You have to know the past to understand the present*". The past and the future are not very different, and this concept is even truer concerning natural hazards. Indeed, if an area has been affected in the past by significant natural events (earthquakes, tsunamis, floods), similar events will certainly occur again in the future, because every natural past event has causes that continue to work in the same way. Likewise, each past event has implications for future ones, and, when a disaster event takes place, society is deeply transformed. As such, for a better understanding of the future, we must study the past with a critical approach.

The designed educational activities described in this paper have two main aims: (1) to demonstrate that a careful study of past earthquakes and tsunamis, on the basis of modern concepts, can offer a present-day understanding in order to infer events that may happen in the future; (2) to trigger critical thinking, stimulating reflection and increasing preparedness in response to earthquakes and tsunamis and thus strengthening resilience [1].

According to the "Hyogo Framework for Action 2005–2015: Building the Resilience of Nations and Communities to Disasters, International Strategy for Disaster Reduction and to the Chart of the Sendai Framework for Disaster Risk Reduction (2015–2030)", particularly

in relation to Priority 1 (Understanding Disaster Risk), we aimed to substantially reduce the disaster risk and loss of lives for a more resilient society (UNISDR, 2015; https://www.preventionweb.net/files/44983_sendaiframeworkchart.pdf, accessed on 12 December 2023). The INGV activity addressed to schools was focused on the prevention of future natural events, favoring the reduction of hazard exposure and the vulnerability to disaster.

Since the introduction of the *Episodes of Situated Learning (ESL)* method in Italy by Prof. Rivoltella [2,3], the INGV has applied it to its research topics, in order to involve students by using technology at school [4].

Following the *ESL* methodology, complex scientific concepts are "divided" into small pieces of fundamental knowledge that students can acquire in order to transfer key concepts to their peers. Students' communication products, e.g., digital posters, are the creative results of their personal learning processes. *ESL* is structured in three phases: preparatory, operative and debriefing, and implementing the principles of *flipped lessons* [5]. Researchers are not mere "dispensers of knowledge", but they are tutors in an assisted laboratory, acquiring significant observations and considerations through shared research and the reworking of *learning by doing* activities.

The need to design engaging and educational distance learning activities to support students and teachers during the COVID-19 pandemic led us to test the online use of this innovative digital teaching method, previously used only in person [6,7].

On the occasions of European Research Night 2020 and on both World Water and World Earth Day 2021, the distance lab activities "Earthquakes: history teaches us the future" and "Tsunamis: history teaches us the future" were performed by the INGV. The first ESL was tested within the NET Science Together project, and the second one was tested both within the European Interreg Italy–Croatia project called PMO-GATE and within the project Future Responsible Citizens (FCR)—Educational Paths of Civil and Environmental Responsibility, in collaboration with the Italian Associazione per lo Sviluppo Sostenibile e Centro di Educazione Ambientale (ASSOCEA Messina APS).

Approximately 2200 students from middle school third-year classes (13-year-old students) and from all classes of high school (14-to-18-year-old students) took part in our *ESL* experimentation. After the INGV researchers' explanatory lesson on past earthquakes and tsunamis, students became "researchers for a day" and prepared, independently, 150 creative digital artifacts describing some of the most important historical events in their region. Researchers' knowledge was at the service of the school, using a curiosity-driven approach in order to help homebound teachers and students during the pandemic. The activities were designed to increase the awareness of the risks related to earthquakes and tsunamis through the study of past events, bringing students closer to the world of research and encouraging them to independently develop content after a discussion with the experts. The main purpose was for them to understand how the past is an important key to reducing the impact of future events. At the end of this experiment, some students reported their experience in "*Noi Magazine*", the insert in *Gazzetta del Sud* dedicated to education.

At the end of each scientific event, different satisfaction questionnaires were distributed to both teachers and students. The feedback was very useful in assessing the perceptions and appreciation of our educational learning activities, and they encouraged us in the design of new *ESL*s. In addition, every teacher, every student, and all classes were handed a certificate of participation for the event.

## 2. Materials and Methods

### 2.1. Episodes of Situated Learning (ESL)

This paper intends to present a new didactic model based on the *ESL* method. *ESL* works on the minimum didactic unit, i.e., the barycenter from which the teacher's didactic action is developed. The structure of an *ESL* consists of a natural ternary that is related to the school setting's management [8]. Each phase can be considered a stage in the learning process that corresponds to a type of teaching logic, as schematized in Figure 1.

**The three phases of ESL** (C. Rivoltella, 2013)

**Preparation/Anticipatory**

| Homework Framework Stimulus | → | Experience Conceptualize Analyze | Problem solving |

**Activity/Operative**

| Delivery | → | Analyze Apply | Learning by doing |

**Debriefing**

| Restructuring | → | Discuss Share | Reflective learning |

**Figure 1.** ESL methodological framework [8].

The three fundamental steps are always presented in sequence, but they are related to each other: (1) the ***anticipatory moment***, which consists of a *stimulus situation* (conceptual framework, video, image, experience, document, testimony); (2) the ***operative moment***, consisting of a *micro-production activity* (analysis/creation of a text starting from a problem to be solved); (3) the ***debriefing moment***, which consists of restructuring what has been realized in the previous two steps [9].

Students design a communication product in a context that challenges their knowledge, skills, attitudes, and competences; on the other hand, teachers enforce learning and, thanks to this method, can evaluate all three phases of *ESL* activity to achieve a formative assessment. Indeed, the students' evaluation is, at the same time, evolutionary (it verifies the development of competences), comprehensive of the *in itinere* evaluation moments, and coherent with the actions carried out [10,11].

The *ESL* method has its origins in *mobile learning* and it has been widespread since the introduction of tablets in schools. It should be considered as an integrated approach to teaching. This model entails the radical redefinition of all three macro-actions into which teaching is divided: (1) *planning*, which is rethought in modular terms [12]; (2) *communication*, where the concept of "frontal lessons vs. active didactics" is overcome, focusing on problem solving and making and sharing new digital products, and ending with a collective reflection; (3) *assessment*, evolving towards the concept of new assessment, with particular attention to embedded and cumulative tasks [13].

The design of an *ESL* requires some fundamentals: selecting the *microcontents*, providing scaffolding, determining and supporting the role of the teacher, and assessing situated learning. This implies careful design work [14] rather than planning, with a perspective that could be defined as "assembling cultural objects" [15,16].

The didactics for *ESL* consist of finding simple solutions to complex phenomena, as exemplified by problem solving in a complex situated context. Applying a "*simplex*" strategy means striving for very advanced solutions, reducing the effort required to manage them. The "*simplexity*" neologism reflects a possible complementary relationship between complexity and simplicity, looking for simplicity through design [17,18]. Thanks to Berthoz's theory of *simplexity*, the *ESL* method is found to be a significant learning experience with a strong cognitive transferability. Classrooms using the *ESL* method apply the "enactivist" concept, which involves the dynamic interaction of students with their environment [19].

The *ESL* has a flipped-up teaching approach (at home, students obtain information; at school, they learn), which is quite different from a traditional lesson (at school, students learn concepts; at home, they study). Homework is for learning and the acquisition of new skills, while classwork is for reworking and understanding [20].

In the *ESL* method, the *flipped classroom* is integrated with *cooperative learning*, allowing one to reach the highest levels of Bloom's taxonomy, which, starting with the basis of Bloom's pyramid, is structured in the following way:

- **remember**: recall facts and basic concepts, which means to define, duplicate, list, memorize, repeat, and state;
- **understand**: explain ideas or concepts, meaning to classify, describe, discuss, explain, locate, recognize, report, select, and translate;
- **apply**: use information in new situations, which means to execute, implement, solve, use, demonstrate, interpret, operate, schedule, and sketch;
- **analyze**: draw connections among ideas, which means to differentiate, organize, relate, compare, contrast, distinguish, examine, experiment, question, and test;
- **evaluate**: justify a stand or decision, which means to appraise, argue, defend, judge, select, support, value, critique, and weigh;
- **create**: produce new or original work, which means to design, assemble, construct, conjecture, develop, formulate, and investigate [21].

Students following the *ESL* method must progress through all the steps to be able to design a communication product that can explain to others the micro-content that they have reworked.

According to the European Commission's Recommendation 2006/962/CE regarding the *Key Competences for Lifelong Learning*, *ESL* is a methodological proposal for a "smart school" that trains students to be able to optimize their resources by developing problem-solving skills [22]. The development of *critical thinking* is favored by the reflection necessary to carry out the assignment. Good critical thinkers are able to split a broad idea into many parts: they can examine each part, question biases, and come to a reasonable conclusion [23].

*2.2. Experiments in Innovative Geosciences Education through ESL as Interactive Teaching Tools for Modern School*

Applying the *ESL* methodology to geosciences requires an innovative system to involve both students and teachers in a new approach to teaching and learning, using simple digital tools to learn even complex concepts. This method is also very successful in activities addressed to schools, aimed at reducing natural hazards and teaching good practices for civil protection purposes. Our first *ESL* experiment produced comics on safe behavior in the case of natural events. Then, within the framework of the *KnowRisk (Know your city, Reduce seISmic risKthrough non-structural elements)* project, the *ESL's* final digital outputs were creative products dedicated to non-structural seismic risk reduction [24–28]. Another *ESL*, titled "*A nuoto tra i vulcani Italiani*" ("*Swimming among the Italian volcanoes*"), developed to raise awareness of the *ESL* method among teachers, was designed to explain Italian seamounts using interactive maps. This latter *ESL* was also used on the occasion of the *European Researchers' Night 2019*, held in person in Pisa within the framework of the *Bright* project.

On the basis of our previous experience, during the COVID-19 pandemic, we created some *ESLs* whose final products were interactive maps to be discussed online during the debriefing phase via *distance learning*. Moreover, since topics related to macroseismic studies had to be dealt with, producing interactive maps as a digital output was the most suitable tool to approach this type of research.

In order to teach students that the past is fundamental to understand the present for a better future, not only in history but also in geophysics, we developed a learning activity that involves encouraging them to work as geophysicists. We therefore used the study of past seismic and tsunamigenic events to engage students in a self-made exercise in which they were involved directly, working in groups, experimenting with new digital techniques, and presenting their results to others as researchers do at scientific conferences [29–31].

2.2.1. "Earthquakes: History Teaches Us the Future—Researchers for a Day with Experimentation in Didactics for ESL"

The first *ESL* designed during the COVID-19 pandemic was "*Earthquakes: history teaches us the future*", addressed to students from middle school third-year classes (ISCDE2) and from all classes of high school (ISCDE3). Within the framework of the *NET Science Together* project (a Marie Curie project funded by the European Commission), the *ESL* activity was performed during a special edition of the *European Researchers' Night 2020*, held exclusively online for schools, in the week of 23–27 November during the COVID-19 pandemic.

The "*Earthquakes: history teaches us the future—researchers for a day with experimentation in didactics for ESL*" special event was structured into two live streaming sessions. During the first one, which occurred on 23 November, the INGV researchers explained the method to the students and teachers and they conducted a short preparatory lesson on the key concepts of earthquakes. In particular, this covered the definition of earthquakes and seismic waves, the distinction between microseismology and macroseismology, the difference between magnitude and intensity, and the tools and safe behaviors to adopt for risk prevention and self-protection. The researchers also underlined the importance of knowing about past earthquakes and they explained how scientists work to reconstruct their effects. This was a fundamental step because the students' task was to choose a historical earthquake of their interest and to study it as if they were true researchers. The students were asked to create an effective communication product addressed to their peers. To steer students in their work, they were provided with video stimulus, a list of earthquakes to choose from, a link to the CPTI15 historical seismicity database [32], and the Mercalli macroseismic scale. These students, working in *cooperative learning*, were divided into different groups in each class. The researcher selected, for each region, a list of the most significant earthquakes from the CFTI15 database. Each group of students, after choosing an earthquake from the list, searched for additional detailed information about the event, assigning a macroseismic intensity value to each locality involved. As a final product, they had to prepare a digital interactive intensity map of the chosen earthquake with linked multimedia files. They had also to provide the seismic history of the studied region, starting from the list of earthquakes supplied by the researchers and integrating it with the data extracted from the CFTI15 database (see Figure 2).

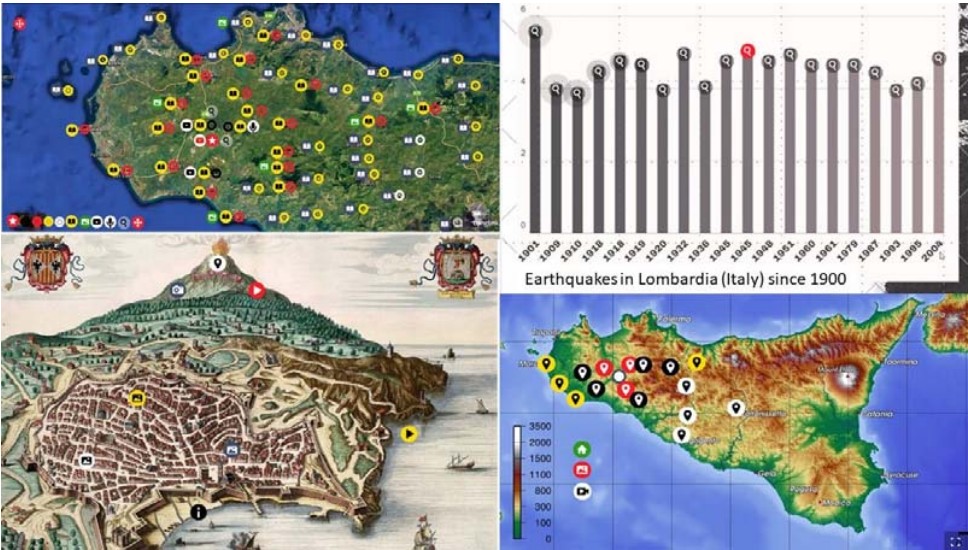

**Figure 2.** Some of the interactive maps designed by students involved in the "*Earthquakes: history teaches us the future—researchers for a day with experimentation in didactics for ESL*" educational online special event. Top left and bottom right: two interactive maps for the reconstruction of the 1968 Belice (Sicily) earthquake. Top right: the seismic history of the Belice (Sicily) area from 1900 to the present day. Bottom left: a historical map of the volcano Etna dating back to the late 1600s, taken by the students as the basis for creating the interactive map of the 1693 Val di Noto (Sicily) earthquake.

Over the following two days, the INGV researchers supported the classes during their creative work phase.

This event was attended by a total of 464 students and 18 teachers from 23 classrooms throughout Italy. In particular, 204 students were from 10 middle school classes, producing 20 macroseismic interactive digital maps, while 260 high school students from 13 classes produced 38 interactive digital macroseismic maps. In one day, the researchers corrected the 58 interactive products, realized by using the *ThingLink* free app (see Table 1).

**Table 1.** Schools involved in the "*Earthquakes: history teaches us the future—researchers for a day with experimentation in didactics for ESL*" educational online experience.

| Region | Middle Schools (ISCDE2) | MS Classes | MS Students | High Schools (ISCDE4) | HS Classes | HS Students | Total n. Students |
|---|---|---|---|---|---|---|---|
| Lazio | 1 | 1 | 28 | 1 | 2 | 53 | 81 |
| Lombardia | 2 | 3 | 57 | - | - | - | 57 |
| Sicilia | - | - | - | 1 | 4 | 83 | 83 |
| Toscana | - | - | - | 1 | 3 | 62 | 62 |
| Liguria | 2 | 2 | 39 | 1 | 1 | 10 | 49 |
| Marche | 1 | 4 | 80 | - | - | - | 80 |
| Umbria | - | - | - | 1 | 3 | 52 | 52 |
| | | | | | | **TOTAL** | **464** |

Each map was the result of the work of a team of 4–5 students via *distance learning.* During the second live streaming session, held on 27 November, the leaders of each student team presented the product of their group for a collective *debriefing* phase conducted by the researchers.

The *Earthquakes: history teaches us the future* ESL is summarized in Table 2.

**Table 2.** The "*Earthquakes: history teaches us the future*" ESL.

| Title of *ESL* | *Earthquake: History Teaches Us the Future* | | |
|---|---|---|---|
| **Authors** | Giovanna Lucia Piangiamore and Alessandra Maramai, Istituto Nazionale di Geofisica e Vulcanologia (INGV) | | |
| **Target (classroom, students' age, etc.)** | Students from third class of middle school (ISCED 3) and first to fifth classes of high school (ISCED 4), dealing with the study of earthquakes | | |
| **Skills that the *ESL* aimed to develop in students** | Control and adaptability/complex dimensions/transferability<br>Ability to represent identified relationships<br>Ability to use groups of information for a personal representation of the topic<br>Ability to use digital tools to visualize the distribution of tsunami effects on the coast<br>Ability to produce information<br>Ability to use technology in a goal-oriented way<br>Ability to produce a creative cultural object (interactive map) | | |
| **What will the student be able to do at the end of this *ESL*?** | The aim was to bring young people closer to the world of research, fostering the personal investigation of the topics discussed with the experts and enabling an understanding of how the past is an important key to reduce the impacts of future earthquakes.<br>Through this experience, the students could understand that macroseismic study is fundamental in seismology and to reconstruct the seismic history of an area is a very complex task. The knowledge of the seismic and tsunamigenic history of a place is the result of meticulous and in-depth work, including the analysis of catalogues and historical documents. The *ESL* was performed at the end of the learning unit on seismic risk. Starting from basic geological–geophysical knowledge (notions of earthquakes, differences between magnitude and intensity, seismic hazards, macroseismic maps), the students learned that earthquakes are not predictable and the past can teach us about the future. | | |
| **PHASES** | **DESIGN** | *App* | **TIMES** |

**Table 2.** *Cont.*

| Title of *ESL* | *Earthquake: History Teaches Us the Future* | | |
|---|---|---|---|
| **PREPARATORY** | **Homework**<br>A stimulus lesson on the study of historical earthquakes was proposed, aimed at understanding the methodology used by seismological researchers to reconstruct past seismic events and to realize a macroseismic map.<br><br>Students were provided with a list of the main earthquakes that had occurred in their territory from 1900 to the present day, the Catalogue of Strong Italian Earthquakes (CFTI15) (http://storing.ingv.it/cfti/cfti5/, accessed on 12 December 2023), the Mercalli macroseismic scale (attached) as a necessary tool for the assignment of intensities, and a very short explanatory video on macroseismic surveys (https://youtu.be/HsDdzy_YOUA?list=PL9AYW9rU1MgBHjM4eis98JGXrO5gxVWYO, accessed on 12 December 2023).<br>Students could derive useful information also from (http://www.blueplanetheart.it/2020/06/ingv-mille-anni-sismicita-italiana-nel-catalogo-cpti-database-macrosismico-dbmi/, accessed on 12 December 2023) and from http://protezionecivile.unionerenolavinosamoggia.bo.it/images/Piano_ProtCiv/Sezioni_Piano_PC/Sezione2/SR4.1_RG001_Terremoto.pdf, https://ingvterremoti.com/i-terremoti-in-italia/, accessed on 12 December 2023).<br><br>**Framework**<br>At school, the teacher described the key concepts of earthquakes with a Powerpoint presentation:<br>- what earthquakes are and why we study historical ones;<br>- the strongest earthquakes in Italy;<br>- the difference between magnitude and intensity;<br>- intensity assessment;<br>- macroseismic maps.<br><br>Students studied these notions.<br><br>**Stimulus**<br>The teacher provided a video stimulus on the topic:<br>https://ingvterremoti.com/2014/12/01/i-terremoti-nella-storia-memoria-condivisa-tradizioni-popolari-e-il-terremoto-del-16-novembre-1894-nella-calabria-meridionale/, accessed on 12 December 2023.<br><br>The aim was to elicit students' curiosity and enthusiasm about the historical seismology of our country, developing an awareness that the Italian territory has high seismicity and that earthquakes recur cyclically. | Youtube (to see the video-stimulus)<br><br><br>Powerpoint (to present the *framework*)<br><br><br>DROPBOX (to insert file)<br><br><br>Notepad (for notes) | Time required for each student to complete homework<br><br><br><br><br><br>5′ |
| **OPERATING PHASE** | **Assignment**<br>As in the attached example, display the earthquakes of your area in the timeline.<br><br>Create with *ThingLink* an interactive intensity map (macroseismic map) of the chosen earthquake, with any geographical base. Consult the CFTI15 catalogue to obtain the information needed to construct the map. Enrich the digital map with additional information on the chosen earthquake (parts of original texts with damage descriptions, historical images, maps, etc.) (you can use the Internet for material).<br><br>*(The final product was realized by small groups of 4–5 students, in order to have, at the end, a collection of interactive maps representative of the Italian territory. They could underline the importance of macroseismic studies to be emphasized in the debriefing).*<br>*Each group of students placed the final product into a folder in Dropbox/Drive so that the shared document allowed for the immediate discussion of the results.* | *ThingLink* (for interactive maps)<br><br>Word processor (for the graph)<br><br><br>DROPBOX/ DRIVE (for the assignment) | 50′<br><br><br><br>15′<br><br><br><br>5′ |
| **DEBRIEFING PHASE** | **Assessment and Discussion**<br>*The teacher* critically analyzed the output of the students, selecting a few intensity maps and asking the students to present them and explain the reasons for their choices; the teacher corrected the final products, made suggestions, and actively participated in the collective discussion, clarifying the appropriate conclusions, highlighting what is most important, and clarifying misconceptions about earthquakes.<br>*The students* analyzed the results and they reflected on their own final products and those of their peers, making observations.<br>Metacognitive thinking was developed, resulting from the discussion with others about their final products and the way in which they were carried out.<br><br>**Output**<br>Corrected digital works could be shared and posted in the Dropbox/Drive folder. | Notepad (to write conclusions)<br><br><br><br>DROPBOX/ DRIVE (to archive the final report) | 30′ |

Each *ESL* must be supplemented by an *assessment rubric*, which is a summary of statements describing a competence, indicating the degree of achievement of the fixed objectives; in addition, a *declination of competences grid* helps teachers in evaluating the students involved in the *ESL* activities. In our case, both the *assessment rubric* and the *declination of competences grid* were the same for both the "*Earthquakes: history teaches us the future*" and the "*Tsunamis: history teaches us the future*" ESLs (see Tables 3 and 4).

*Assessment rubric*

**Table 3.** The "*Earthquakes: history teaches us the future*" and the "*Tsunamis: history teaches us the future*" assessment rubric.

| SIZES | LEVELS | | | |
|---|---|---|---|---|
| | Partial | Essential | Medium | Excellent |
| **Interpreting the representation** | Interprets representations only when guided, has difficulty in extrapolating information and in identifying its overall meaning; has difficulty in using different codes and/or switching from one language to another | Interprets representations in an essential manner, partially extrapolates information and identifies some significant aspects, manages to use different codes and/or switches from one language to another | Autonomously interprets representations, extrapolates the most important information by identifying the meaning and reworking it using different codes and/or switching from one language to another in an appropriate manner | Interprets with confidence representations, extrapolates the most important information by identifying hidden meanings and reworking it using different codes and/or fully switching from one language to another and in a personal manner |
| **Acting in an organizational and emotional-relational autonomy** | Cannot act autonomously depending on the situation, needs support to overcome difficulties | Can act semi-autonomously according to the situation and should be encouraged to make the right choices (has some insecurities) | Can act autonomously and correctly, adapting to different situations | Is able to act autonomously, appropriately and consciously with confidence, adapting to different situations without losing heart. He/she is an example for others and supports peers in difficulty |
| **PRODUCE** | Uses technology in a simple way and can only produce simple composition, if guided | Uses technology in an adequate manner and produces less than satisfactory work | Makes appropriate use of technology and produces simple but correct work | Makes targeted use of technology and produces original and personal work |

*Declination of competences grid*

**Table 4.** The "*Earthquakes: history teaches us the future*" and the "*Tsunamis: history teaches us the future*" declination of competences grid.

| SKILLS (among the 8 'Key' Competences) | SIZES (Qualifying Aspects) | CRITERIA (What the Student Must Be Able to Do) | MARKERS (Objective Evidence) |
|---|---|---|---|
| **Scientific skills** | KNOWING THE REPRESENTATION | Knows how to navigate between the various types of representation | Knows the various representations and their structural characteristics |
| **COMMUNICATE** | KNOWING THE NECESSARY PROCEDURES TO INTERPRET REPRESENTATION | Knows how to proceed in reading the representation | Knows the phases of reading and identifies the knowledge/skills required to do it |
| | INTERPRETING THE REPRESENTATION | Can extrapolate information from the representation | Explains a representation by identifying its global and analytical meaning |
| **Social and civic skills ACT AUTONOMOUSLY AND RESPONSIBLY** | Organizational autonomy | Can manage time, space and materials | Knows and sets up the necessary tools for various school activities, carries out individual and/or group work in the required time according to purpose Recognizes and respects rules |
| | Emotional-relational autonomy | Knows how to respect others, collaborate, help, listen and participate in discussions | Follows the rules of the classroom (how to participate in collective phases, waiting for their turn, respecting the times and working methods of their classmates) |
| **Digital skills** | COLLECT | Knows how to find information | Identifies the most reliable sources |
| | | | Critically selects the necessary information |
| | ORGANIZE | Can link information | Uses technology in a targeted manner |
| | PRODUCE | Can produce information | Uses technology in a targeted manner |
| | | | Produces a creative cultural object |

A further area in which we experimented in the "*Earthquakes: history teaches us the future—researchers for a day with experimentation in didactics for ESL*" event was a *Path for Transversal Skills and Orientation (PTSO)* pathway in a Scientific Lyceum in Pavia, within the framework of "*Taramelli's Year*". The project involved 34 students, from the fourth and fifth classes, who were divided into eight working groups, participating with great interest. Six 2-h online meetings with the students were held by INGV researchers in collaboration with the University of Pavia. Regarding the other *ESL*s, during the first meeting, the method was explained and the key concepts of earthquakes were introduced. Each group chose an event to be studied and had to produce a digital macroseismic map of the selected earthquake. The other online meetings were used to check the state of the art of the activity and to clarify students' doubts. Meanwhile, since the COVID-19 pandemic restrictions had been reduced, it was decided that each working group would present the results of their work to students of other classes in their school, during a ceremony held in person in the Lyceum's auditorium in May 2022.

Another initiative was carried out within the framework of the "*Taramellian Year*", organized by the University of Pavia in collaboration with the INGV. In this activity, the learners were teachers, attending a 10-h training course titled "*Teaching for ESL—Earthquake: history teaches us*". The course was developed in four meetings held online from April to May 2022, and a specific lecture on the *ESL* method was held by Prof. Rivoltella.

### 2.2.2. Tsunamis: History Teaches Us the Future—Researchers for a Day with Experimentation in Didactics for ESL

The enthusiastic feedback received from both the teachers and students participating in "*Earthquakes: history teaches us the future—researchers for a day with experimentation in didactics for ESL*" led us to design a similar *ESL* dealing with tsunamis: "*Tsunamis: history teaches us the future—researchers for a day with experimentation in didactics for ESL*". This online learning activity was aimed at reconstructing past tsunamis within the framework of the *PMO-GATE* project, in which the INGV is involved in the study of tsunamis and meteotsunamis for the prevention of natural hazards, and within the framework of the *FCR* project, in which the INGV is involved in *CON.I.RI. (CONvivere con I Rischi naturali—Living with natural hazards)*. This latter collaboration led to an interaction with the daily newspaper *La Gazzetta del Sud*—in particular, with its supplement, *Noi Magazine*, edited by high school students from Calabria and Sicily. This is why we decided to involve only middle and high schools from Sicily and Calabria in this *ESL* experiment.

On the basis of our experience in the *ESL* on historical earthquakes, we gave students more time to design their digital maps, exploiting the opportunities provided by two important events: World Water Day (22nd March) and World Earth Day (22nd April). The first live streaming session was held by the INGV researchers on the occasion of World Water Day 2021, to explain the *ESL* method and to raise awareness about tsunamis, a natural phenomenon whose risk on the Italian coast is very often underestimated. At the same time, during the lesson, the fundamentals of tsunamis were introduced: the definition of tsunamis, why and how past tsunamis are studied, the tsunami intensity scale, and the most relevant tsunamis in the Mediterranean Sea and, particularly, along the Italian coast. In addition, it was explained how to reconstruct the tsunamigenic history of a locality/area of interest.

The *ESL* activity was focused on the tsunami following the 1908 Messina (Sicily, Italy) earthquake, the strongest that has ever occurred in Italy. For this event, the students had to reconstruct the effects produced in the territory near their school.

"*Tsunamis: history teaches us the future—researchers for a day with experimentation in didactics for ESL*" was attended by a total of 1707 students and 43 teachers from 107 classrooms in Sicily and Calabria. In particular, approximately 900 students were from 60 classes in middle school, producing 75 macroseismic interactive digital maps, while, among all high school classes, approximately 800 students from 47 classes produced 90 interactive digital macroseismic maps (see Figure 3).

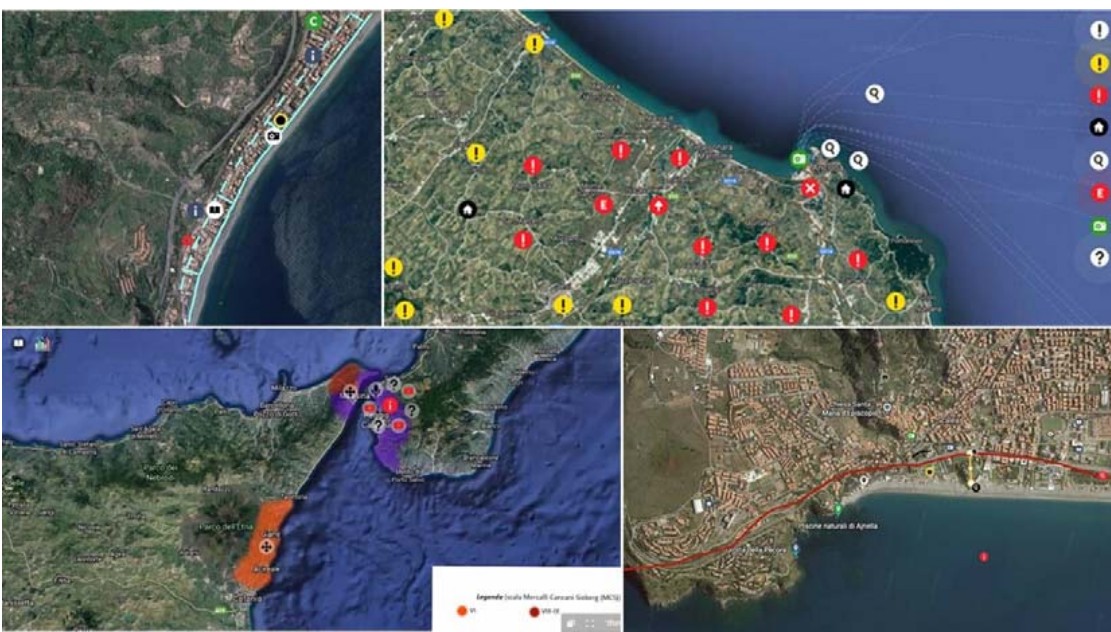

**Figure 3.** Some of the interactive maps designed by students involved in the "*Tsunamis: history teaches us the future—researchers for a day with experimentation in didactics for ESL*" educational online special event.

In one week, the researchers corrected the 165 interactive maps, realized by using the *ThingLink* free app (see Table 5).

**Table 5.** Schools involved in the "*Tsunamis: history teaches us the future—researchers for a day with experimentation in didactics for ESL*" educational online special event.

| Province | Middle Schools ISCDE2 | MS Classes | MS Students | High Schools ISCDE3 | HS Classes | HS Students | Total n. Students |
|---|---|---|---|---|---|---|---|
| Caltanissetta | 1 | 3 | 55 | - | - | - | 55 |
| Catania | 8 | 17 | 270 | 4 | 10 | 128 | 398 |
| Cosenza | 5 | 16 | 144 | | | | 144 |
| Siracusa | 2 | 8 | 120 | 2 | 30 | 523 | 643 |
| Trapani | 1 | 3 | 54 | - | - | - | 54 |
| Reggio Calabria | 2 | 9 | 150 | - | - | - | 150 |
| Messina | 1 | 2 | 50 | | | | 50 |
| Enna | 1 | 2 | 45 | | | | 45 |
| Crotone | - | - | - | 1 | 2 | 39 | 39 |
| Vibo Valentia | - | - | - | 1 | 5 | 129 | 129 |
| | | | | | | **TOTAL** | **1707** |

Moreover, for this activity, the classrooms were divided into working groups of 4–5 students, assisted remotely on demand by the INGV researchers.

The *Tsunamis: history teaches us the future ESL* scheme is summarized in Table 6.

Due to the large number of participants in the special event concerning the earthquake *ESL* described in Section 2.2.1, it was difficult to manage the timing of the final live debriefing day. Indeed, it was necessary to give voice to all 58 groups of students who had presented their products to their peers and discussed them with the researchers.

**Table 6.** The "*Tsunamis: history teaches us the future*" ESL.

| *ESL* Title | *Tsunamis: History Teaches Us the Future* | | |
|---|---|---|---|
| Authors | Giovanna Lucia Piangiamore and Alessandra Maramai, Istituto Nazionale di Geofisica e Vulcanologia (INGV) | | |
| Target (classroom, students' age, etc.) | Students from third class of middle school (ISCED 3) and first to fifth classes of high school (ISCED 4), dealing with the study of tsunamis | | |
| Skills that the *ESL* aims to develop in students | Control and adaptability/complex dimensions/transferability<br>Ability to represent identified relationships<br>Ability to use groups of information for a personal representation of the topic<br>Ability to use digital tools to visualize the distribution of a tsunami's effects on the coast<br>Ability to produce information<br>Ability to use technology in a goal-oriented way<br>Ability to produce a creative cultural object (interactive map) | | |
| What will the student be able to do at the end of this EAS? | The aim was to bring young people closer to the world of research, fostering a personal investigation of the topics discussed with the experts and enabling an understanding of how the past is an important key to reduce the impacts of future tsunamis.<br>Through this experience, the students could understand that tsunamis are closely related to earthquakes and that, even for tsunamis, events that occurred in the past can be repeated in the future with similar characteristics. Reconstructing the effects of tsunamis is often difficult because tsunami damage is added to that caused by the generating earthquake. The knowledge of the seismic and tsunamigenic history of a place is the result of meticulous and in-depth work, including the analysis of catalogues and historical documents. In particular, the case study of the tsunami associated with the Messina earthquake of 1908 was analyzed.<br>The *ESL* was performed at the end of the learning unit on tsunami risk. Starting from basic geological–geophysical knowledge (notions of earthquakes and tsunamis, seismic and tsunami hazard maps, tsunami intensity assessment), the students learned that, even for tsunamis, the past can teach us about the future and that tsunami warning systems exist for risk reduction. | | |
| PHASES | DESIGN | *App* | TIMES |
| PREPARATORY PHASE | **Homework**<br>A stimulus lesson on the study of historical tsunamis was proposed, aimed at understanding the phenomenon and how researchers reconstruct its effects. The activity mainly focused on the reconstruction of the tsunami following the Messina earthquake of 1908.<br>The database of the effects of Italian tsunamis was provided (https://tsunamiarchive.ingv.it/en/tsunami-catalogues/ited-italian-tsunami-effects-database, accessed on 12 December 2023), and also the Ambraseys-Sieberg scale as a tool for the assignment of tsunami intensities (attached). Students had to derive useful general information on tsunamis from videos, https://www.ted.com/talks/alex_gendler_how_tsunamis_work/transcript?language=it#t-201827, accessed on 12 December 2023; https://www.youtube.com/watch?v=qTd62yuSOQM, accessed on 12th December 2023; from surveys of post-event effects, https://vimeo.com/51246302 (accessed on 12 December 2023); and from the INGV Tsunami Warning Centre website, https://programming14-20.italy-croatia.eu/web/pmo-gate, accessed on 12 December 2023; https://www.ingv.it/ricerca/progetti-e-convenzioni/progetti/pmo-gate#abstract-2, https://cat.ingv.it/en/, accessed on 12 December 2023.<br><br>**Framework**<br>At school, the teacher described the key concepts of tsunamis with a Powerpoint presentation:<br>- what tsunamis are and why we study historical ones;<br>- the strongest tsunamis in the world and in Italy;<br>- the Messina tsunami of 1908;<br>- tsunami intensity assessment;<br>- a map with the distribution of tsunami effects.<br>Students studied these notions.<br>**Stimulus**<br>The teacher provided a video stimulus on the 1908 Messina earthquake as an in-depth study:<br>https://www.youtube.com/watch?v=KkKAUY5IUVI, accessed on 12 December 2023 and<br>https://www.youtube.com/watch?v=1pPGSylKLW8, accessed on 12 December 2023.<br><br>The aim was to teach students that, even in our country, tsunamis represent a real, often underestimated risk for coastlines. | Youtube (to see the video-stimolo)<br><br><br>Power point (to present the *framework*)<br><br><br><br>Notepad (for notes) | Time required for each student to complete homework<br><br>5′ |

**Table 6.** *Cont.*

| ESL Title | Tsunamis: History Teaches Us the Future | | |
|---|---|---|---|
| **OPERATING PHASE** | **Assignment**<br>Create with ThingLink, with any geographical basis (Google Maps or similar is suggested), an interactive map representing the areas with the greatest tsunami risk on the coasts of Sicily and Calabria, starting with the data of the 1908 Messina tsunami. You should identify and highlight the "strategic infrastructures" (schools, hospitals, police stations, etc.) present in the study area today. You have to consult the EMTC2.0/ITED online database to get the information needed to realize the map. Enrich your digital work with additional information on the tsunami for the different locations (description of effects, historical images, videos, etc.). You can use the Internet to search for material.<br><br>Visualize, for the chosen locations, the tsunamigenic history, redrawing it and inserting it into your map, correlated with the relevant information obtained from the tsunami intensity scale provided.<br><br>*(The final product was realized by small groups of 4–5 students, in order to have, at the end, a collection of interactive maps representative of the entire Sicilian and Calabrian coast. They could underline the importance of the tsunami phenomenon in the area, to be emphasized in the debriefing).*<br>*Each group of students placed the final product into a folder in Drive so that the shared document allowed for the immediate discussion of the results.* | *ThingLink* (for interactive maps)<br><br><br>Word processor (for the tsunamigenic history diagram)<br><br><br>DRIVE (for the final product) | 60′<br><br><br><br><br>15′<br><br><br><br><br><br>5′ |
| **DEBRIEFING PHASE** | **Assessment and Discussion**<br>*The teacher* critically analyzed the output of the students, selecting a few tsunamigenic maps and asking the students to present them, explaining the reasons for their choices; the teacher corrected the final products, made suggestions, and actively participated in the collective discussion, clarifying the appropriate conclusions, highlighting what was most important, and clarifying misconceptions about tsunamis.<br>*The students* analyzed the results and they reflected on their own final products and those of their peers, making observations on the different products.<br>Metacognitive thinking was developed, resulting from the discussion with others about their final products and the way in which it was carried out.<br><br>**Output**<br>Corrected digital works could be shared and posted in the Drive folder. | Notepad (to write conclusions)<br><br><br><br>DRIVE (to archive the final report) | 30′ |

In response to the huge number of participants in the tsunami *ESL*, the number of sessions for the second live streaming event on the occasion of World Earth Day 2021 was doubled, separating the students from the 60 middle school classes and the students from the 47 high school classes. Therefore, the INGV researchers conducted two second live streaming events dedicated to the *debriefing* phase on 22nd April 2021, in which the researchers and students discussed the results obtained for a constructive comparison and exchange of experiences among the participating classes.

## 3. Results

In the context of fast socio-cultural transformation, teaching must adapt, embracing the new educational needs through new means of engagement and learning towards a positive outcome. This need was particularly evident during the COVID-19 pandemic, when it was necessary to combine the traditional in-person classroom method with *e-learning* [33]. Some activities that usually took place in the classroom were performed online by means of mobile systems (smartphones, tablets). Indeed, technology played a very important role, not only in supporting in-person lessons but also in favoring interaction remotely and in providing online resources [34]. In this particular historical period, it was necessary to explore

new means of using digital teaching methodologies, such as *ESL*. This method, based on the *flipped class* model, has a teaching approach that focuses on emotional, cognitive, and behavioral components. The students autonomously acquire new information at home; then, with their teachers and schoolmates, they rework, share, and discuss their assignments [35]. Our experience in using the *ESL* method applied to geosciences at schools highlighted that this approach is an effective tool to enhance motivation and learning, developing positive emotions and favoring higher levels of self-efficacy [36,37]. This is an opportunity to train students' skills in an active and participative environment. The low level of perceived anxiety in students also improves their learning [38]. Indeed, this method works in a *real-life* context, in which learning occurs in everyday situations and not only in dedicated teaching environments. The didactic becomes more experiential and reflective, providing meaningful learning. This context improves the teacher–student relationship and grants teachers the necessary conditions to achieve an effective and authentic assessment, observing students during all three phases of *ESL* [39].

To celebrate 10 years of the method, the Catholic University of Brescia organized the "*ESL* Day", titled "Gli *EAS* tra didattica e pedagogia di scuola—10 anni di metodo" ("*ESL* between didactic and school pedagogy—10 years of the method"). On this occasion, we were invited by Prof. Rivoltella to present a lecture on our *ESL* experiments in the earth science disciplines, as we were pioneers in the application of the method in teaching geosciences and geophysics.

The sharing of the experiences of students from across Italy, during a period in which the entire population was homebound due to the COVID-19 pandemic, was the main strength of the activity.

Concerning "*Earthquakes: history teaches us the future—researchers for a day with experimentation in didactics for ESL*", we collected many warm and enthusiastic comments of gratitude from the students as they felt part of a community, able to learn in an active and cooperative manner. Among the many testimonies of satisfaction, the most notable ones from high school students were as follows.

"An interesting and useful project on a subject about which people are often misinformed and underestimate the risks. It gave us the opportunity to learn how to reduce natural hazards".

"We were able to learn new and interesting topics, and at the same time we learnt how to work together in the most effective and efficient way".

"This activity allowed us to know earthquakes in a different way from the scholastic one, certainly more engaging, giving us a creative stimulus. I was fascinated to learn how a historical seismologist works".

"This project was really interesting both because it was different from all the others, and because of its practical approach. We used new digital tools to manage earthquake data working in teams. It was fun and made the work less burdensome, even dealing with a very delicate subject. I hope to do similar activities soon".

In addition, numerous appreciative e-mails were received from the teachers involved in the activity. Similarly, this positive trend was highlighted by the answers to the questionnaires distributed to the students and teachers after the learning activity.

Moreover, regarding "*Tsunamis: history teaches us the future—researchers for a day with experimentation in didactics for ESL*", we collected many very appreciative posts from the participating schools and reports in *Noi Magazine*, a supplement of *La Gazzetta del Sud* dedicated to education. In the related TGweb, the students reported about their experiences like "little journalists". The following are two of the most significant comments received from teachers:

"It was a stimulating and engaging initiative that let the students be protagonists and induced them to ask questions and look for answers, as researchers do".

"A fruitful and exciting activity that puts students at the center of their learning process, making them the protagonists of their own education. An opportunity to experience a conscious and responsible use of new technologies".

At the end of each *ESL* activity, a survey was conducted by providing a satisfaction questionnaire, which was different for teachers and students. The feedback was very useful in assessing the perceptions and appreciation of our educational proposal and they guided us in the design of new learning activities. The answers received, in fact, confirmed the appreciation and interest of both students and teachers. The following figures (Figures 4–9) show the results of the satisfaction questionnaires administered.

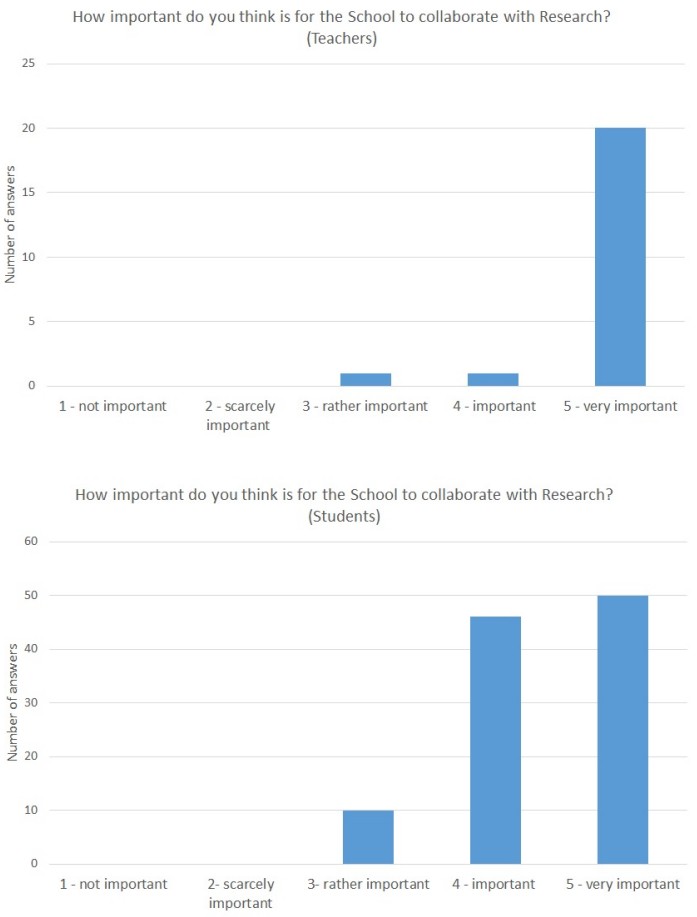

**Figure 4.** Cumulative values showing percentage of feedback received for the first question from teachers (**top**) and students (**bottom**), concerning the *ESL* experimentation. The x-axis denotes the degree of appreciation. The y-axis denotes the number of teachers/students who completed the questionnaire.

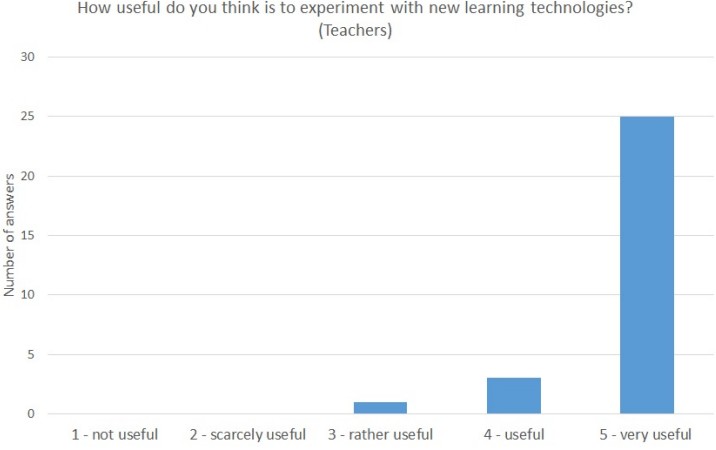

**Figure 5.** *Cont.*

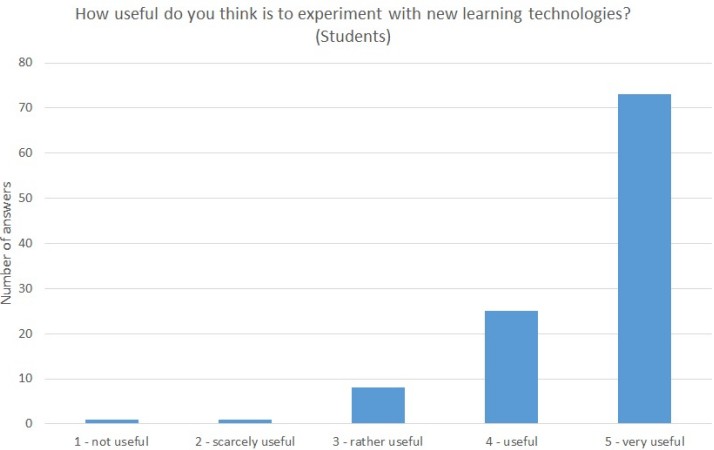

**Figure 5.** Cumulative values showing percentage of feedback received for the second question from teachers (**top**) and students (**bottom**), concerning the *ESL* experimentation. The x-axis denotes the degree of appreciation. The y-axis denotes the number of teachers/students who completed the questionnaire.

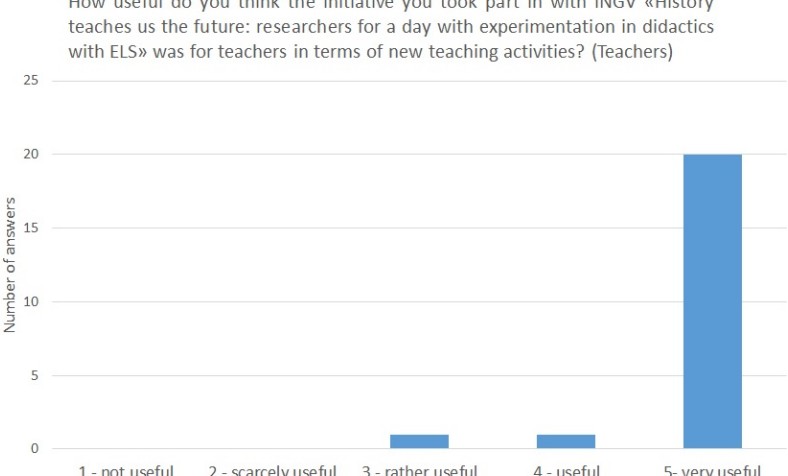

**Figure 6.** Cumulative values showing percentage of feedback received for the third question from teachers, concerning the *ESL* experimentation. The x-axis denotes the degree of appreciation. The y-axis denotes the number of teachers/students who completed the questionnaire.

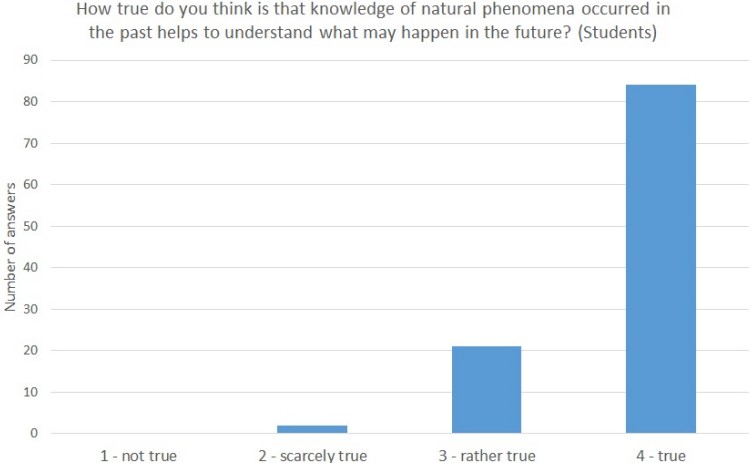

**Figure 7.** Cumulative values showing percentage of feedback received for the third question from students, concerning the *ESL* experimentation.

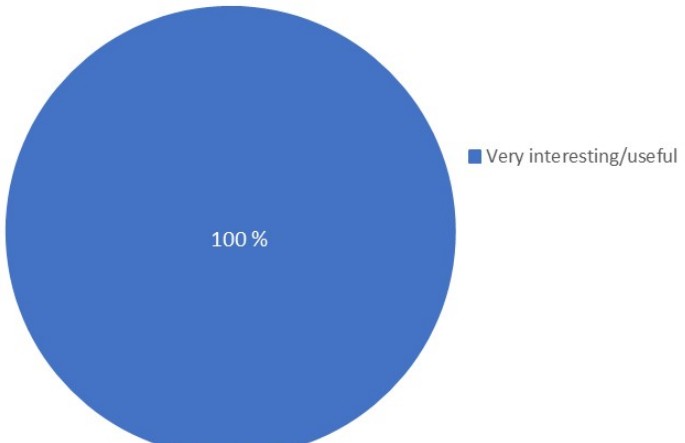

**Figure 8.** Pie chart showing cumulative percentage of feedback received for the fourth question from teachers, concerning the *ESL* experimentation.

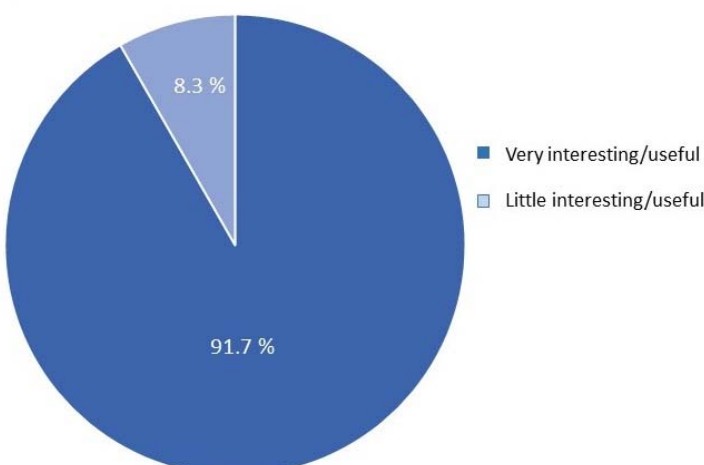

**Figure 9.** Pie chart showing cumulative percentage of feedback received for the fourth question from students, concerning the *ESL* experimentation.

Concerning the importance of the collaboration between schools and researchers, in general, the teachers answered the questionnaire with a higher degree of positivity than the students. The students seemed to be less impressed by the projects than the teachers, but if we combine the answers describing the initiatives as "very interesting" with those describing them as "interesting", this accounts for 87 percent of the respondents. Nonetheless, twenty years of experience with this type of initiative in schools has shown that students generally do not enjoy completing satisfaction questionnaires at the end of an activity. In fact, much less feedback is typically collected compared to the number of participating students.

The *ESL*s tested during the COVID-19 pandemic allowed us to reach a very high number of students and teachers simultaneously connected on the internet, improving the interaction between schools and researchers during this difficult period. The encouraging feedback received on the satisfaction questionnaires from all participants proves the educational efficacy of the designed *ESL*s.

Within the framework of the *90th Congress of the Italian Geological Society "Geology Without Borders"*, held online in September 2021, a talk titled *"Experiments of e-learning: ESL (Episodes of Situated Learning) during the Pandemic"* was presented. As a result of this presentation, the University of Pavia permitted us to experiment with didactics for *ESL* within the *Path for Transversal Skills and Orientation (PTSO)*. The activity was conducted through distance learning during the first two phases, while the debriefing phase was held in person. The resulting *blended* (or hybrid) *learning* was an integrated teacher (in this case, researcher)/learner approach strategy in a combination of different learning environments [40].

Future developments include the design of new *ESL*s on scientific topics, both in the field of earth sciences and geophysics and in environmental issues, such as one realized by request for the *Geothermix 2023* conference. This *ESL* dealt with geothermics in Italy and a *ThingLink* interactive map was presented within the keynote speech titled "Experiments in innovative Geosciences Education through Episodes of Situated Learning (EAS) as interactive teaching tools for modern School: the case of Geothermal and Geo-resources", presented at the *Geothermix 2023* conference held in Pisa in November 2023.

## 4. Discussion

An *ESL* is a "simplex" strategy, creating connections between *real life* and didactics, helping students to find simple tools to aid their learning and cope with complexity and to develop students' skills by means of devices [41]. The term "simplexity" originates from the biological strategies through which living species adapt themselves to the surrounding complexity. The solutions to deal with different situations consider past experiences and anticipate future ones; these are new means of addressing problems so that actions may be taken quickly and efficiently [42]. The philosophy behind *ESL* is "say a lot in a few words and, if possible, make people think more than they say" [43], triggering a process to simplify various principles for a complex world.

According to Howard Gardner's *theory of multiple intelligences* [44,45], humans do not have only one intellectual capacity, but they have many types of intelligence that fulfill eight criteria: visual–spatial, linguistic–verbal, logical–mathematical, body–kinesthetic, musical, interpersonal, intrapersonal, and naturalistic. Many teachers use multiple intelligences in their teaching to integrate Gardner's theory into the classroom. In relation to this, one of the strengths of the *ESL* method is the development of *active learning* in which students are free to express all their abilities [46]. Another of the most relevant benefits of the *ESL* method is that it allows the student to use his/her own intelligence with respect for all identities [47]. Each student might demonstrate specific strengths and abilities, so that, in a classroom, a wide range of different talents is available [48]. These are the reasons for which *ESL* is an appealing didactic method that could be performed worldwide to favor a modern school approach.

*ESL* is a methodology that is particularly suitable for the study of natural hazards in order to promote the spread of good practices, fostering the development of *emotional intelligence*, which is key to well-being, when the right and left parts of the brain are in equilibrium. Daniel Goleman describes *emotional intelligence* as an ability to understand and manage one's own emotions and influence the emotions of others [49]. Examining the interpersonal and intrapersonal aspects, human intelligence is deeply linked to the social and affective dimensions of human life. *ESL* is a didactic methodology that is able to put into practice this creativity found between rational and emotional thinking. Thanks to this creativity, students can learn spontaneously in a friendly environment. The advantages of flipping are to promote peer interaction and collaboration skills, to make learning central rather than teaching, to foster independent learning, to encourage stronger student engagement, and to provide increased individualized attention [50].

The *ESL*s developed during online meetings with the INGV researchers (*informal learning*) focused on *microlearning* activities to achieve self-production, starting from *real life* (knowing the seismological setting of the student's own area, which is a crucial factor

in seismic hazard reduction). This is placed within the context of gathering together knowledge, skills, attitudes, and competences. Students learn more effectively if they start from situations of daily life or in their territory, in which they can focus their attention, acting based on *microcontent* and developing their *thinking skills*, *problem solving*, and *reflective learning* [51]. The situated education action as a minimum but significant unit is a clear example of how it is possible for teachers to work with *ESL* in the virtual classroom with students at home, connected by the internet and not only in person [52].

## 5. Conclusions

*ESL* proposes an innovative means of studying with the use of new technologies, as a new method of teaching and learning. Our experience with *ESL* with 13–18-year-old students shows that a good *lesson plan* for this method can foster the development of *critical thinking*. The activities are designed to support authentic language use and to develop a type of thinking that involves making fair, careful judgements and evaluations based on evidence, reason, reflection, and open-mindedness. In brief, ESL involves teaching learners to analyze complex settings [53]. Students have to understand the core problem and suggest solutions through the creative assembly of cultural objects [54]. In order to create a digital communication product for their peers, the students must acquire the ability to focus on the key concepts of the studied topic, reworking and understanding them in a more comprehensive manner. An example is the *ESL* named "*Earthquakes: history teaches us the future: researchers for a day with experimentation in didactics for ESL*", in which the students discovered the differences between *microseismology* (analysis of seismic signals) and *macroseismology* (damage estimation). Under the guidance of the researchers, the students understood how multi-faceted seismologists' work is: it can deal with different branches of seismology, studying the various aspects of earthquakes for a greater understanding of the Earth system as a whole. During the second phase of the *ESL*, to enrich the final macroseismic map with further information about the chosen earthquake, the students searched the internet and analyzed parts of original texts describing the earthquake's effects, as well as images and historical maps. In this way, they were able to identify themselves in the work of the historical seismologist, performing an engaging educational activity.

The *ESL* model leads to higher levels of scholastic engagement in students and reduces their levels of perceived anxiety. Moreover, this method facilitates embedded assessment activities, thanks to assignments and exercises focused on a particular learning outcome. Indeed, the evaluation is formative: it is not only a score for the students' final digital creative product, but it is the result of the teacher's observation during all three *ESL* phases. The assessment encompasses the behavior of the student, who, independently and in groups, works on a project and then presents its product to others. The teacher can evaluate the student's work, using an *assessment rubric*, designed as an ad hoc summary statement describing their competences and indicating the degree of achievement of the set objects. The synthetic outlines of the *declination of competences grid* provide a further detailed evaluative tool that forms an objective evaluation criterion, pointing out which parameters are to be considered in the evaluation. This type of evaluation allows teachers to identify students' skills within a particular domain (e.g., social, academic), favoring future *lesson plan*s to support each student's progress [55]. Moreover, it enables them to discover whether the designed learning activity is effective. *ESL* needs a carefully designed *lesson plan* to create situated and meaningful learning experiences, leading students to realize digital artifacts and fostering the personal appropriation of content. The effort of finding suitable video stimuli and learning trigger activities requires a lot of time and sometimes discourages teachers, who have to search the internet to design creative activities through free apps, addressing students' self-production activities [56]. Therefore, teachers particularly appreciate the availability of ready-to-use *ESL*s on topics of their interest to implement in their classes. In this case, the teachers do not need to have extensive

knowledge of the method or particular expertise to apply it. They only have to follow the exercise step by step.

The *ESL* experiments of the series "*History teaches us the future: researchers for a day with experimentation in didactics for ESL*" are two examples of the digital teaching of geoscience, created to help students and teachers during the COVID-19 pandemic. They are perfectly aligned with the INGV's mission to distribute geophysics research results. This activity is not merely dissemination and communication: it is education [57]. It is also a means to bring future responsible citizens closer to researchers' work and to make students aware that public research is at the service of society. Researchers and schools can achieve a great deal together, focusing on protection, knowledge, and awareness of natural disaster prevention, and they can encourage good practices and safe and sustainable behavior [58–60]. This is a means to build resilience at school, involving students, teachers, school leaders, and families too, because children can be a valuable vehicle by which to increase awareness among adults [61].

**Author Contributions:** Conceptualization, methodology, software, validation, formal analysis, investigation, resources, data curation, writing—original draft preparation, writing—review and editing, visualization, supervision, project administration, G.L.P. and A.M. All authors have read and agreed to the published version of the manuscript.

**Funding:** This research received no external funding.

**Data Availability Statement:** Data are contained within the article.

**Acknowledgments:** The authors thank Anna De Santis for the technical support during the online special events. They also thank Claudia Lupi for the collaboration within the *Taramellian Year* and to Pier Cesare Rivoltella for enhancing our geoscience and geophysics experimentation using *ESL.* The first author thanks Monica De Vecchi for coaching and revealing to her the great potential of the *ESL* method by inspiring new designs. The authors are also grateful to Irene Rosati Valdambrini for the English language revision.

**Conflicts of Interest:** The authors declare no conflicts of interest.

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
