# Peer review of "When the Past Teaches the Future: Earthquake and Tsunami Risk Reduction through Episodes of Situated Learning (ESL)"

_geosciences, doi:10.3390/geosciences14030065_

Round 1
Reviewer 1 Report
Comments and Suggestions for Authors
Review to Manuscript ID: geosciences-2803306
When the past teaches the future: earthquake and tsunami risk 2 reduction through Episodes of Situated Learning (ESL)
By: Giovanna Lucia Piangiamore and Alessandra Maramai
Piangiamore and Maramai present two Episodes of Situated Learning (ESL) case studies, that were focused on learning what past earthquakes and tsunamis can teach us in order to improve response and resilience of our society, had such natural hazards impact again in the future. The authors, who are affiliated with INGV, Italy, where these ESLs were developed, guided and trained about 2200 Italian Middle and High schools students and their teachers. The students and their teachers found much interest and importance in performing the ESLs, and expressed high enthusiasm in learning how acquiring knowledge of natural hazards that occurred in the past can help them understanding and preparing for such events in the future.
General Comments
The opening paragraph in the Introduction is very important in placing the motto of this project, yet it needs, in my opinion, wider perspective. At the background of the “past teaches the future” concept stands the uniformitarianism principle, formulated at the time by the geologists James Hutton and Charles Lyell. They summarized their understandings by the well-known slogan: ”The Present Is the Key to the Past”. This perception should also be presented and discussed, especially in a paper that deals with principles of scientific education within the geosciences discipline. In fact, study of past accounts of earthquakes and tsunamis is conducted with much care and extrapolated onto the future on the base of nowadays concepts, scientific models and modern professional terminology. Thus, there are ongoing discussions and constructive dialectics between past historical accounts and present-day understandings in order to infer what can happen in the future.
Does ESL approach is performed elsewhere than in Italy? How ESL corresponds with other worldwide educational methods? Can the Italian ESL taken to other places?
Does teachers and/or students need some preliminary background, expertise, skills, etc., before experiencing the described ESLs?
Please present the age of the students also in the introduction, especially for readers who are not familiar with the Italian educational system.
Following lines 227-229: is there a specific methodology how the student need to present their scientific results deliveries? In other words, does the ESL provides instructions how to make and present scientific results, such as structure of a common geoscience abstracts, maps, figures, talks, posters, etc.?
Lines 349 – 387 are kind of self-appreciation of the project – which BTW, does deserve it. However, the questionnaires present the appreciation of the teachers and student in a better balanced way, rather than a set of selected superlatives chosen by the authors. I would suggest focusing on the analysis of the questionnaire results because it demonstrates the full, unbiased spectrum of opinions and appreciation, which is anyhow are very good. The above mentioned lines can be moved to an appendix.
I noticed that in general, the teachers were more decisive in their opinion and answered the questionnaire with higher positive degrees (mainly bin 5) than the students (mainly bins 3-5). In other words, the students seemed to be less impressed from the projects than the teachers. Is that so, and why?
Also, I wonder whether there were any differences by gender in replying the questionnaire.
The discussion is hard to follow, the two paragraphs are quiet long and heavy. Consider simplifying the messages, dividing and heading subsections, etc.
Figures
Figure 1: What is “EAS”?
Figures 2 and 3: explain what are the four panels mean? Is there a need to credit the students who created these maps?
Figures 4-7: explain what are the 5 bins (degree of appreciation?), and the vertical axes (number of teachers/students?)?
Tables
Table 2: Row of “Skills that the ESL…” should regard earthquake rather than tsunamis?
Please rephrase: “Students should try to derive useful information from…”
Table 4: “Can produce informazioni”: I believe you meant “… information”?
Technical Comments
The titles “Earthquakes/Tsunamis: history teaches us the future…” repeat many times. Consider (not necessary) acronym it (e.g. E-htu, T-htu, whatever you like) for fluent reading.
Lines 315-318: Does this paragraph belongs to section 2.2.1?
Lines 327-329: please rephrase the first sentence.
Line 346: “contest” or “context”?
Author Contributions
Consider compacting the list and simplifying the reading by rephrasing: Conceptualization, methodology, software, … etc: G.L.P. and A.M.; and: writing—review and editing, A.M. and G.L.P.
References
Is there a reference to the key note talks during the GeothermiX 2023, and “Geology Without Borders” during the 90th Congress of the Italian Geological Society held online in September 2021?
The following site link appears twice: https://www.preventionweb.net/files/44983_sendaiframeworkchart.pdf
Comments on the Quality of English LanguageA few were already listed above
Author Response
Does ESL approach is performed elsewhere than in Italy? How ESL corresponds with other worldwide educational methods? Can the Italian ESL taken to other places?
Some insertion in the text has been done.
Does teachers and/or students need some preliminary background, expertise, skills, etc., before experiencing the described ESLs?
Some insertion in the text has been done.
Please present the age of the students also in the introduction, especially for readers who are not familiar with the Italian educational system.
Some insertion in the text has been done.
Following lines 227-229: is there a specific methodology how the student need to present their scientific results deliveries? In other words, does the ESL provides instructions how to make and present scientific results, such as structure of a common geoscience abstracts, maps, figures, talks, posters, etc.?
No, the ESL method does not supply any instruction to students in order to present their digital product. You can imagine a situation like a congress, in which every group, represented by a leader, shows its digital output to all the other participants. Every talk is followed by a general discussion conducted by the researcher or teacher.
Lines 349 – 387 are kind of self-appreciation of the project – which BTW, does deserve it. However, the questionnaires present the appreciation of the teachers and student in a better balanced way, rather than a set of selected superlatives chosen by the authors. I would suggest focusing on the analysis of the questionnaire results because it demonstrates the full, unbiased spectrum of opinions and appreciation, which is anyhow are very good. The above mentioned lines can be moved to an appendix.
Thanks for your suggestion. Anyhow, we do not consider appropriate to move the students' comments into an appendix because they are the result of a spontaneous demonstration of gratitude towards an initiative that actively involved them in a very special historical period, during which they were isolated at home because of the Pandemic. Moreover, twenty years of experience with this type of initiatives with schools showed that students generally do not like to fill out the satisfaction questionnaires at the end of the activity. In fact, we always gather less feedback than the number of the participating students. In addition, we have repeatedly found that the most listless students in each class randomly answer questions almost as if they were doing a stunt.
I noticed that in general, the teachers were more decisive in their opinion and answered the questionnaire with higher positive degrees (mainly bin 5) than the students (mainly bins 3-5). In other words, the students seemed to be less impressed from the projects than the teachers. Is that so, and why?
Some insertion in the text has been done.
Also, I wonder whether there were any differences by gender in replying the questionnaire.
Deliberately no gender differentiation was made.
The discussion is hard to follow, the two paragraphs are quiet long and heavy. Consider simplifying the messages, dividing and heading subsections, etc.
We accepted your suggestion and according to the guidelines of Geoscience Journal, we inserted in the text the “Conclusion” paragraph (“Conclusions: This section is not mandatory but can be added to the manuscript if the discussion is unusually long or complex. https://www.mdpi.com/journal/geosciences/instructions), simplifying the discussion paragraph.
Figures
Figure 1: What is “EAS”? The figure has been replaced.
Figures 2 and 3: explain what are the four panels mean? Some insertion in the text has been done.
Is there a need to credit the students who created these maps?
It is not necessary to credit the students who made the maps shown in the examples in the figures as this was group work for the special event. Each group was identified by a number and not by the names of its members.
Figures 4-7: explain what are the 5 bins (degree of appreciation?), and the vertical axes (number of teachers/students?)?
Some insertion in the text has been done.
Tables
Table 2: Row of “Skills that the ESL…” should regard earthquake rather than tsunamis?
In the caption of Table 2 is written: Table 2. The “Earthquakes: history teaches us the future” ESL. This means that is referred to earthquakes.
Please rephrase: “Students should try to derive useful information from…” done
Table 4: “Can produce informazioni”: I believe you meant “… information”? done
Technical Comments
The titles “Earthquakes/Tsunamis: history teaches us the future…” repeat many times. Consider (not necessary) acronym it (e.g. E-htu, T-htu, whatever you like) for fluent reading.
Thank you for the suggestion but we prefer not to use an acronym that could create confusion with teachers, that are our stakeholders.
Lines 315-318: Does this paragraph belongs to section 2.2.1?
Some insertion in the text has been done.
Lines 327-329: please rephrase the first sentence.
Why??
Line 346: “contest” or “context”?
It was a refuse, thanks. Done
Author Contributions
Consider compacting the list and simplifying the reading by rephrasing: Conceptualization, methodology, software, … etc: G.L.P. and A.M.; and: writing—review and editing, A.M. and G.L.P.
Done
References
Is there a reference to the key note talks during the GeothermiX 2023, and “Geology Without Borders” during the 90th Congress of the Italian Geological Society held online in September 2021?
Concerning Geothermix 2023 there is no reference because it was presented as invited speech at the Geothermix conference held in Pisa on November 2023 (http://geothermix2023.dst.unipi.it/index.php/en/program/keynotes)
As regards the 90th Congress of the Italian Geological Society “Geology Without Borders” held online in September 2021, the talk titled “Experiments of e-learning: ELS (Episodes of Situated Learning) during the Pandemic” is in the book of abstract:https://www.geoscienze.org/files/download/trieste%202021/Abstract%2090mo%20Congresso%20SGI_DEF.pdf. And the content is in the paper you are reviewing.
The following site link appears twice: https://www.preventionweb.net/files/44983_sendaiframeworkchart.pdf
Ok, removed
Reviewer 2 Report
Comments and Suggestions for Authors
The manuscript by Piangiamore and Maramai emphasizes the importance of learning from the past to face the future more consciously, highlighting the key role of schools in disseminating knowledge on natural phenomena and promoting behavior change. It discusses how the INGV engages students in active learning activities, particularly during online scientific events in the pandemic, providing tools and experiences to prepare for earthquakes and tsunamis while fostering interest in historical seismic studies.
Although this article is very meaningful, I believe it deviates from the theme of the journal. It would be more suitable for Sustainability or Education Sciences. Below are my comments.
(1) In the Introduction, I am interested in remote laboratory activities. In the implementation process, how did researchers interact with students through online teaching methods, encouraging them to independently prepare creative digital artifacts at home? Furthermore, how did this unique teaching approach provide support for education during the pandemic?
(2) The ESL method is the core of this article. How is the ESL method related to mobile learning, and how has it become widespread since the introduction of tablets in schools? How is this method considered as an integrated approach in teaching?
(3) How does the integration of flipped classroom and cooperative learning in the ESL method contribute to achieving higher levels of learning outcomes for students within the cognitive levels of Bloom's taxonomy?
(4) As researchers, how do you guide students to integrate the list of earthquakes with data extracted from the CFTI15 database?
(5) In Section 2.2.2, is there any involvement in the study of non-seismogenic tsunamis in the research on tsunamis? These tsunamis may be triggered by volcanic eruptions or landslides, and they are equally significant.
(6) Is the survey for feedback too simple, with only two choices, "very interesting" and "little interesting," lacking intermediate options?
Author Response
The manuscript by Piangiamore and Maramai emphasizes the importance of learning from the past to face the future more consciously, highlighting the key role of schools in disseminating knowledge on natural phenomena and promoting behavior change. It discusses how the INGV engages students in active learning activities, particularly during online scientific events in the pandemic, providing tools and experiences to prepare for earthquakes and tsunamis while fostering interest in historical seismic studies.
Although this article is very meaningful, I believe it deviates from the theme of the journal. It would be more suitable for Sustainability or Education Sciences. Below are my comments.
Our paper was written following a request from the Editor. We also had this doubt and when we wrote the cover letter, we expressed it. We asked them to let us know in a short time whether we should submit it or refer us to other journals. In the end, the Editor accepted the work and sent it to the reviewers.
- In the Introduction, I am interested in remote laboratory activities. In the implementation process, how did researchers interact with students through online teaching methods, encouraging them to independently prepare creative digital artifacts at home? Furthermore, how did this unique teaching approach provide support for education during the pandemic?
The researchers give input to the creative activity by introducing the basic concepts that are fundamental tools to carry out the activity, conveying the passion for their research to the students and explaining the relevance of Research for Society. In addition, the researchers are available to clarify doubts during the operative phase. At the end, after the correction of the digital elaborates, researchers, as conveners, chair the student’s presentations. The activity concludes with a collegial discussion (debriefing).
The ESL method, based on active learning in which the student is the protagonist of his own learning, was particularly well suited during the Pandemic, at a time when high school students in Italy were isolated at home for a very long time.
- The ESL method is the core of this article. How is the ESL method related to mobile learning, and how has it become widespread since the introduction of tablets in schools? How is this method considered as an integrated approach in teaching?
The ESL method was born in Italy in 2013 along with the introduction of digital devices (mainly tablets) at school. At that time, some schools considered '2.0' were pioneers and benefited greatly from the introduction of this method at school. That found its greatest spread when the use of tablets and smartphones in schools was more massive. This required a readjustment of teaching methodologies.
How is this method considered as an integrated approach in teaching?
Many studies have shown that blended learning methods, where both face-to-face and distance learning, have proven to be particularly effective in various situations such as the Pandemic.
- How does the integration of flipped classroom and cooperative learning in the ESL method contribute to achieving higher levels of learning outcomes for students within the cognitive levels of Bloom's taxonomy?
As is well known, Bloom's taxonomy describes the learning steps by representing them as a pyramid: at the base there is 'remember' , as the first stage of learning, followed by 'understand', then 'apply', 'analyse', 'evaluate' and at the apex of the pyramid is 'create'. Therefore, the students following the ESL method must progress through all the steps in order to be able to design a communication product that can explain to others the micro-content he has made his own and reworked.
Some insertion in the text has been done.
- As researchers, how do you guide students to integrate the list of earthquakes with data extracted from the CFTI15 database?
The researcher selects, for each region, a list of the most significant earthquakes from the CFT15 database. Each group of students, after choosing an earthquake of their interest from the list, search for additional detailed information about the event, allowing them to assign a macroseismic intensity value to each locality involved by the selected earthquake.
Some insertion in the text has been done
- In Section 2.2.2, is there any involvement in the study of non-seismogenic tsunamis in the research on tsunamis? These tsunamis may be triggered by volcanic eruptions or landslides, and they are equally significant.
In the case study described in Section 2.2.2., the tsunami following the 1908 Messina earthquake 1908 (a seismically induced tsunami) was considered, because a lot of information were available. However, this does not exclude the possibility of designing ESLs concerning tsunamis of non-seismic origin.The only relevant condition is that informations on the effects produced along the coastline are available in the literature.
- Is the survey for feedback too simple, with only two choices, "very interesting" and "little interesting," lacking intermediate options?
For each question, teachers/students have the possibility to select five degree of appreciation. From zero to five.
Some insertion in the text has been done
We would like to thank the anonymous reviewer for her/his precious suggestions that allowed us to improve the manuscript, particularly in the Introduction and Results/Discussion paragraphs.
Reviewer 3 Report
Comments and Suggestions for Authors
Thank you for report! Such activities are also useful for popularizing science and research in general.
Author Response
Comments and Suggestions for Authors
Thank you for report! Such activities are also useful for popularizing science and research in general.
We would like to thank the anonymous reviewer for her/his appreciation of our work.
Round 2
Reviewer 2 Report
Comments and Suggestions for Authors
I thank the authors for their response to my feedback. Thematically, this article is valuable as it elucidates the key role of schools in disseminating knowledge on natural phenomena and promoting behavior change. The author has revised the content according to my suggestions. I am satisfied with the revision.
While I still believe that it leans more towards the social sciences and might not be the best fit for this journal, the author mentioned that it was a commissioned piece by the editor. In this case, I don't object to the publication of this article if the editor agrees.